# Precise *let-7* expression levels balance organ regeneration against tumor suppression

Linwei Wu[1,2,3†], Liem H Nguyen[1,3†], Kejin Zhou[4], T Yvanka de Soysa[5,6,7,8], Lin Li[1,3], Jason B Miller[4], Jianmin Tian[9], Joseph Locker[9], Shuyuan Zhang[1,3], Gen Shinoda[5,6,7,8], Marc T Seligson[5,6,7,8], Lauren R Zeitels[3,10], Asha Acharya[3,10], Sam C Wang[1,3,11], Joshua T Mendell[3,10], Xiaoshun He[2], Jinsuke Nishino[12], Sean J Morrison[12], Daniel J Siegwart[4], George Q Daley[5,6,7,8], Ng Shyh-Chang[5,6,7,8,13], Hao Zhu[1,3*]

[1]Children's Research Institute, Departments of Pediatrics and Internal Medicine, Simmons Comprehensive Cancer Center, University of Texas Southwestern Medical Center, Dallas, United States; [2]Organ Transplant Center, The First Affiliated Hospital of Sun Yat-Sen University, Guangzhou, China; [3]Center for Regenerative Science and Medicine, University of Texas Southwestern Medical Center, Dallas, United States; [4]Simmons Comprehensive Cancer Center, Department of Biochemistry, University of Texas Southwestern Medical Center, Dallas, United States; [5]Stem Cell Transplantation Program, Division of Pediatric Hematology/Oncology, Boston Children's Hospital and Dana Farber Cancer Institute, Boston, United States; [6]Harvard Stem Cell Institute, Harvard University, Boston, United States; [7]Department of Biological Chemistry and Molecular Pharmacology, Harvard Medical School, Boston, United States; [8]The Manton Center for Orphan Disease Research, Howard Hughes Medical Institute, Boston Children's Hospital, Boston, United States; [9]Department of Pathology, University of Pittsburg, Pittsburg, United States; [10]Department of Molecular Biology, University of Texas Southwestern Medical Center, Dallas, United States; [11]Department of Surgery, University of Texas Southwestern Medical Center, Dallas, United States; [12]Howard Hughes Medical Institute, Children's Research Institute, Department of Pediatrics, University of Texas Southwestern Medical Center, Dallas, United States; [13]Stem cell and Regenerative Biology, Genome Institute of Singapore, Singapore, Singapore

*For correspondence: Hao.Zhu@utsouthwestern.edu

†These authors contributed equally to this work

**Abstract** The in vivo roles for even the most intensely studied microRNAs remain poorly defined. Here, analysis of mouse models revealed that *let-7*, a large and ancient microRNA family, performs tumor suppressive roles at the expense of regeneration. Too little or too much *let-7* resulted in compromised protection against cancer or tissue damage, respectively. Modest *let-7* overexpression abrogated *MYC*-driven liver cancer by antagonizing multiple *let-7* sensitive oncogenes. However, the same level of overexpression blocked liver regeneration, while *let-7* deletion enhanced it, demonstrating that distinct *let-7* levels can mediate desirable phenotypes. *let-7* dependent regeneration phenotypes resulted from influences on the insulin-PI3K-mTOR pathway. We found that chronic high-dose *let-7* overexpression caused liver damage and degeneration, paradoxically leading to tumorigenesis. These dose-dependent roles for *let-7* in tissue repair and tumorigenesis rationalize the tight regulation of this microRNA in development, and have important implications for *let-7* based therapeutics.

**eLife digest** The development of animals is guided by the expression of certain genes at critical moments. Many different mechanisms control development; in one of them, the expression of genes can be decreased by molecules called microRNAs. In particular, the group of microRNAs called *let-7* has been intensively studied in roundworms and fruit flies. Although mammals have extremely similar *let-7* microRNAs they seem to be more important during adulthood.

Previous studies using cells grown in the laboratory have shown that mammalian *let-7* microRNAs decrease cell proliferation and cell growth. Furthermore, in mouse models of various cancers, *let-7* microRNAs often reduce tumour growth when they are supplied to adult mice. Therefore, overall the *let-7* group has been classified as genes that act to suppress tumors, and thus protect mice (and most likely humans too) from cancers. However, in-depth analysis of *let-7* microRNAs was still missing.

Wu and Nguyen et al. have now studied mice with liver cancer using strains where they were able to regulate the levels of *let-7*. These mice overproduce a strong cancer-inducing gene in the liver; half were used as controls and the other half were further engineered to have moderately elevated levels of *let-7* expression. Most of the control mice got large cancerous tumors, but only a few mice in the other group developed cancers and the tumors were smaller. This confirmed that *let-7* hinders tumor formation.

Wu and Nguyen et al. also observed that the protected mice were less able to regenerate their liver tissues. Further experiments showed that deleting just two out of ten *let-7* microRNAs enhanced the mice's ability to regenerate liver tissue after injury. These findings indicate that *let-7* microRNAs slow down the growth of both cancerous and normal cells. Lastly, when *let-7* levels were raised to very high levels for a prolonged amount of time this actually led to liver damage and subsequent tumor formation.

This last observation may have important consequences for possible cancer therapies. Some scientists have shown that providing extra *let-7* can slow or even reverse tumour growth, but the findings here clearly point out that too much *let-7* could actually worsen the situation. Since the *let-7* family comprises a handful of microRNAs in mammals, in the future it will also be important to find out to what extent these molecules play overlapping roles and how much they differ.

## Introduction

MicroRNAs are thought to control cellular responses to stresses such as tissue damage and transformation (*Leung and Sharp, 2010*; *Chivukula et al., 2014*), but the impact of this idea is unclear because microRNAs have been understudied in vivo. *let-7* is one of the most ancient and omnipresent microRNAs, yet relatively little is known about its functional roles in mammalian development and physiology. *let-7* was first identified as a gene that regulates the timing of developmental milestones in a *C. elegans* screen (*Reinhart et al., 2000*). In mammals, mature *let-7* is undetectable in early embryos and embryonic stem cells, but becomes highly expressed in most adult tissues (*Schulman et al., 2005*; *Thomson et al., 2006*). A handful of previous studies have implicated *let-7* in body size regulation, metabolism, stem cell self-renewal, and colon carcinogenesis (*Zhu et al., 2011*; *Frost and Olson, 2011*; *Shyh-Chang, et al., 2013*; *Nishino et al., 2013*; *Madison, et al., 2013*, but the core functions of *let-7* in regeneration and disease remain incompletely understood.

In addition to questions about what *let-7* does, it is unknown why so many *let-7s* are expressed at such high levels. In mice and humans, the *let-7* family is comprised of 10 to 12 members who are thought to share a common set of mRNA targets. It has been thought that deep redundancy might make it difficult to discern any phenotypes that individual *let-7s* might have. Essential unanswered questions regarding *let-7* biology include whether *let-7* members are redundant, have unique functions, or are regulated to maintain a specific total dose. Our previous study of Lin28a, which inhibits the biogenesis of each *let-7* member similarly (*Heo et al., 2008*; *Nam et al., 2011*), suggests that total *let-7* dose alterations, rather than regulation of specific members, is important. In transgenic

mice, modest increase in *Lin28a* and consequent 40% suppression of total *let-7* levels promote increased glucose uptake and an overgrowth syndrome (*Zhu et al., 2010*).

In this study we examined the consequences of *let-7* dose disruption in cancer and organ regeneration in genetic mouse models. While *let-7s* have been implicated as a tumor suppressor, this has predominantly been shown in cell lines and xenograft assays (*Guo et al., 2006*; *Chang et al., 2009*; *Iliopoulos et al., 2009*; *Viswanathan et al., 2009*; *Wang et al., 2010*; *Lan et al., 2011*), as well as using exogenous *let-7* delivery to mouse cancer models (*Esquela-Kerscher et al., 2008*; *Trang et al., 2010*; *Trang et al., 2011*). Here, we confirmed the tumor suppressor activity of an endogenous transgenic *let-7* in a *MYC*-driven hepatoblastoma model. However, we found that this same level of *let-7* overexpression impaired liver regeneration after partial hepatectomy (PHx). Furthermore, chronic high-dose *let-7* resulted in severe liver damage and paradoxical liver cancer development. Overall, we provide in vivo evidence that *let-7* expression levels have been developmentally constrained to balance the need for regenerative proliferation against the need to antagonize malignant proliferation, findings with implications for *let-7* based therapies.

## Results

### *let-7g* inhibits the development of *MYC*-driven hepatoblastoma

To study the effect of *let-7* on carcinogenesis, we employed an inducible *MYC*-driven hepatoblastoma model (*Shachaf et al., 2004*). In this model, most *let-7s* are suppressed by more than 60% (*Nguyen et al., 2014*). However, *MYC* affects the expression of many other microRNAs (*Chang et al., 2009*; *Kota et al., 2009*). To test if *let-7* suppression is specifically required for *MYC*'s oncogenic program, we simultaneously overexpressed *let-7g* and *MYC* using a triple transgenic, liver-specific, tet-off model (*Figure 1A*: *LAP-tTA*; *TRE-MYC*; *TRE-let-7S21L* transgenic mice). This transgenic form of *let-7g* is an engineered *let-7* species called *let-7S21L* (*let-7g* Stem + *miR-21* Loop) (*Zhu et al., 2011*), in which the precursor microRNA loop derives from *mir-21* and cannot be bound and inhibited by Lin28b (*Figure 1B*), which is highly expressed in *MYC*-driven tumors (*Chang et al., 2009*; *Nguyen et al., 2014*).

Induction of *MYC* with or without *let-7S21L* was initiated at 14 days of age (*Figure 1C*). By 90 days of age, large multifocal tumors had formed in 88% of the *MYC* alone group, whereas single small tumors appeared in only 27% the *MYC + let-7S21L* group (*Figure 1D–F*) and overall survival was dramatically improved (*Figure 1G*). The level of *let-7g* was increased more than eightfold in both non-tumor and tumor tissues (*Figure 1H*). Tumors from both groups showed similar histology (*Figure 1—figure supplement 1*) and *MYC* expression (*Figure 1I*). Gene-expression within tumors showed that previously validated *let-7* targets involved in proliferation and growth including *Cdc25a* (*Johnson et al., 2007*), *Cdc34* (*Legesse-Miller et al., 2009*), *E2f2* (*Dong et al., 2010*), *E2f5* (*Kropp et al., 2015*), and *Map4k4* (*Tan et al., 2015*) were upregulated in *MYC*-tumors, but suppressed back down to normal levels in the context of *let-7* overexpression (*Figure 1J–L*), suggesting that the repression of these targets restrains *MYC*-dependent tumorigenesis. These data indicated that *let-7g* has potent tumor suppressor activity in the context of *MYC*-driven hepatoblastoma.

### *let-7g* overexpression inhibits liver regeneration after partial hepatectomy

Since increasing *let-7g* was extremely effective at suppressing hepatoblastoma without compromising overall health, we asked if this increase in levels would impact tissue homestasis. We examined *let-7g* overexpression in the setting of liver injuries that drive rapid proliferation and growth. After PHx, *let-7s* in regenerating tissues fell, but returned to normal after forty hours (*Figure 2A*), findings consistent with a previous report (*Chen et al., 2011*). Similarly, *let-7s* also declined acutely after chemical injury with the xenobiotic TCPOBOP (1,4-bis-[2-[3,5-dichloropyridyloxy]] benzene) (*Figure 2—figure supplement 1A*). This shows that while *let-7* increases in a temporally defined fashion during development (*Figure 2—figure supplement 1B*), it can transiently fluctuate after environmental perturbations. To test if the observed *let-7* suppression is necessary for regeneration, we induced *let-7g* in *LAP-let-7S21L* mice and performed PHx (*Figure 2B–D*). The body weight (*Figure 2—figure supplement 1C*), liver function (*Figure 2—figure supplement 1D*), resected liver mass (*Figure 2E*) and histology (*Figure 2—figure supplement 1E*) were unaffected in *LAP-let-7S21L*

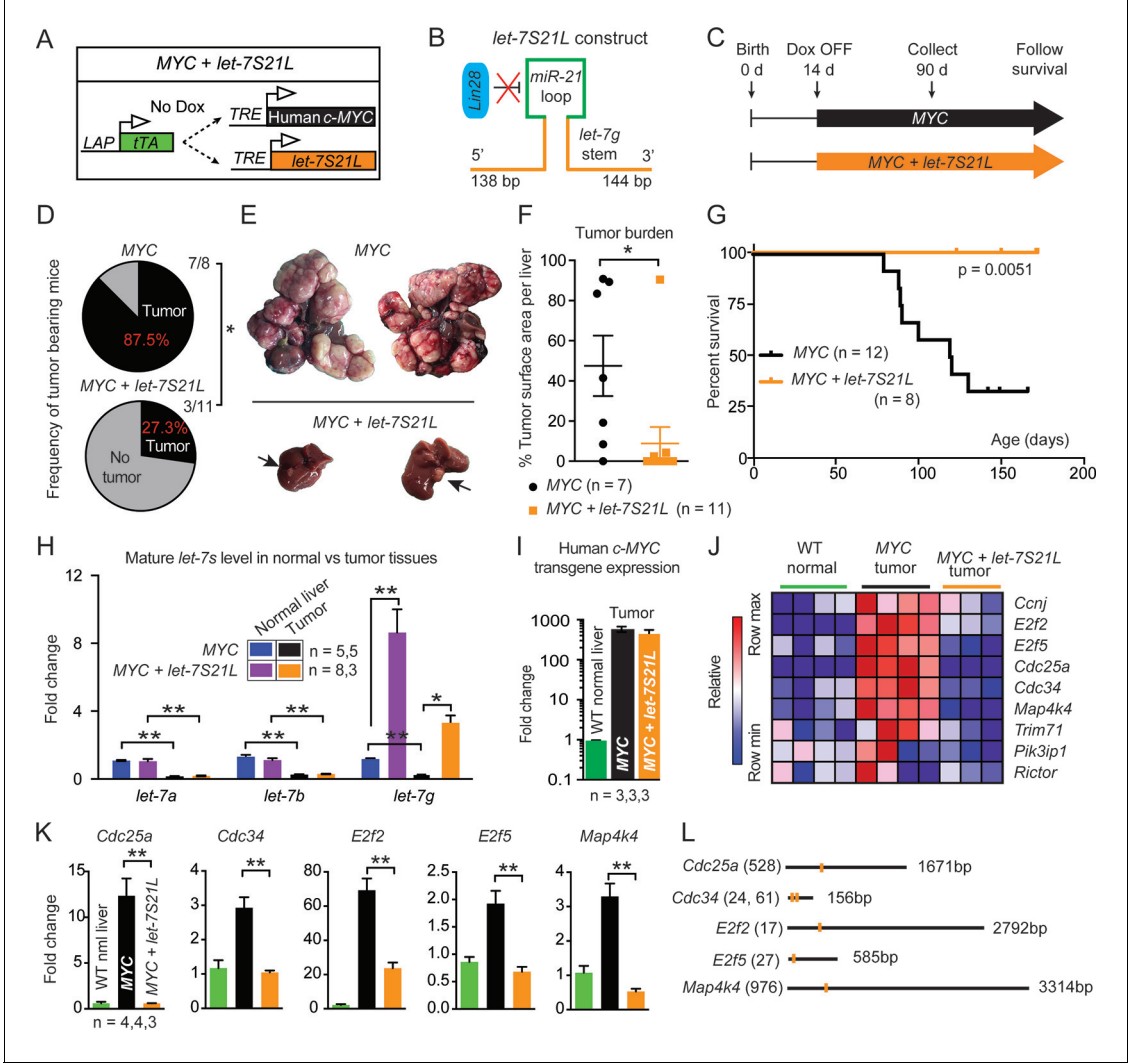

**Figure 1.** *let-7g* inhibits the development of *MYC*-driven hepatoblastoma. (A) Schema of the liver-specific inducible *LAP-MYC +/- let-7S21L* cancer model. (B) *let-7S21L* is a chimeric construct containing the *let-7g* stem, *miR-21* loop, and *let-7g* flanking sequences. (C) Schema showing that *LAP-MYC +/- let-7S21L* mice were induced at 14 days of age, tissues were collected at 90 days of age, and survival was followed. (D) Ninety-day old mice bearing tumors in the *LAP-MYC* (87.5%, 7/8) and *LAP-MYC + let-7S21L* (27.3%, 3/11) mouse models. (E) Livers showing tumors from the above mice. (F) Liver surface area occupied by tumor. (G) Kaplan-Meier curve of *LAP-MYC* alone and *LAP-MYC + let-7S21L* mice. (H) Mature *let-7* expression levels in as determined by RT-qPCR. (I) Human *c-MYC* mRNA expressionin tumors as determined by RT-qPCR. (J) Heat map of *let-7* target gene expression in WT normal livers, *MYC* tumors, and *MYC + let-7S21L* tumors as measured by RT-qPCR. Red is higher and blue is lower relative mRNA expression. (K) Gene expression plotted as bar graphs to show relative changes. (L) Evolutionarily conserved *let-7* target sites within 3'UTRs (Targetscan.org). All data in this figure are represented as mean ± SEM. *p < 0.05, **p < 0.01.

The following figure supplement is available for figure 1:

**Figure supplement 1.** H&E staining of *LAP-MYC* and *LAP-MYC + let-7S21L* tumor-adjacent normal tissues and tumor tissues.

mice compared to control mice. Forty hours after PHx, there was reduced liver mass and decreased Ki-67 in *LAP-let-7S21L* mice (*Figure 2F–H*). Liver mass was no different at four and fourteen days, indicating a kinetic delay but not a permanent impairment (*Figure 2—figure supplement 1F,G*).

To rule out increasing demands on microRNA biogenesis machinery as a mechanism of proliferative suppression, we delivered mature control or *let-7g* microRNA mimics (0.5 mg/kg) into wild-type mice two days prior to hepatectomy using C12-200 lipidoid nanoparticles (LNPs) (*Love et al., 2010*). *let-7g*, but not control mimics, inhibited regeneration (*Figure 2I–K*). While *let-7*

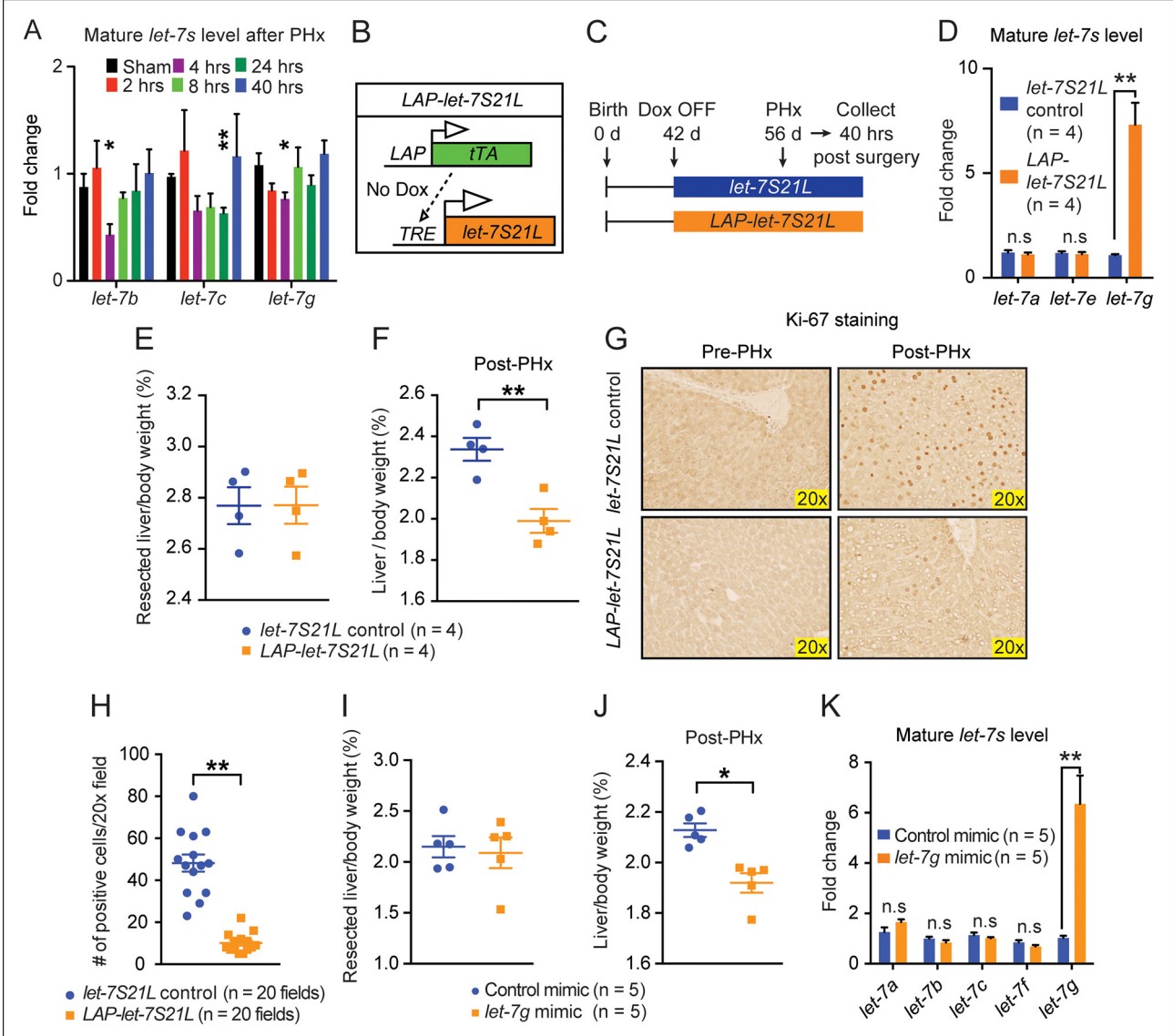

**Figure 2.** *let-7g* overexpression inhibits liver regeneration after partial hepatectomy. (**A**) Mature endogenous *let-7* expression levels in WT C57Bl/6 mice at different time points after PHx as determined by RT-qPCR (n=4 and 4 for each time point). (**B**) Schema of the *LAP-let-7S21L* dox-inducible model. *LAP-tTA* single transgenic mice served as the controls. (**C**) Schema showing that *let-7S21L* control and *LAP-let-7S21L* mice were induced at 42 days of age, PHx was performed after 14 days of induction, and tissues were collected 40 hr post PHx. (**D**) Mature *let-7* expression levels in *let-7S21L* and *LAP-let-7S21L* livers after 14 days of induction (n=4 and 4). (**E**) Resected liver/body weight ratios of *LAP-tTA* Control and *LAP-let-7S21L* mice at the time of PHx (n=4 and 4). (**F**) Liver/body weight ratios 40 hr after PHx (n=4 and 4). (**G**) Ki-67 staining on resected and post-PHx liver tissues. (**H**) Quantification of Ki-67-positive cells (n=2 and 2 mice; ten 20x fields for each mouse were quantified). (**I**) Resected liver/body weight ratios 2 days after intravenous injection of 0.5 mg/kg negative control or *let-7g* microRNA mimics packaged in C12-200 LNPs (n=5 and 5). (**J**) Liver/body weight ratios 40 hr after PHx (n=4 and 4). (**K**) Mature *let-7g* expression levels in mimic treated livers (n=5 and 5). All data in this figure are represented as mean ± SEM. *p<0.05, **p<0.01.

The following figure supplement is available for figure 2:

**Figure supplement 1.** Data associated with *Figure 2*.

overexpression blocked *MYC*-induced tumorigenesis, these data show that a similar increase in *let-7* levels inhibited post-injury organ growth and regeneration.

## Loss of *let-7b* and *let-7c2* is sufficient to enhance liver regeneration

To assess the physiological relevance of our gain-of-function experiments, we examined knockout mice to determine if *let-7* is a bona fide regeneration suppressor. Both *let-7b* and *let-7c2* were conditionally deleted from the liver by crossing *Albumin-Cre* into a *let-7b/c2* floxed mouse ("*let-7b/c2* LKO" mice, *Figure 3A*). Small RNA-sequencing data from Xie et al. showed that *let-7* is one of the most highly expressed microRNA families in the liver and that *let-7b* and *let-7c2* together comprise approximately 18% of the *let-7* total (*Figure 3B*) (*Xie et al., 2012*). Thus, *let-7b/c2* LKO mice have substantial, but far from a complete reduction of total *let-7* levels.

These LKO mice were healthy and showed normal liver/body weight ratios and histology at baseline (*Figure 3—figure supplement 1A,B*). An identical amount of liver mass was resected from *let-7b/c2*$^{Fl/Fl}$ control and *let-7b/c2* LKO mice (*Figure 3C,D*), but LKO mice exhibited significant increases in liver mass and proliferation 40 hr after surgery (*Figure 3E–G*). Four and seven days after PHx, there were no differences in liver weights, indicating that other phases of regeneration were unaffected (*Figure 3—figure supplement 1C,D*). At fourteen days, the liver weight precisely achieved pre-surgery levels in control and LKO mice, indicating accelerated but not excessive regeneration (*Figure 3—figure supplement 1C,D*). There was no compensatory upregulation of other *let-7s* in pre- or post-PHx tissues (*Figure 3H*), supporting the concept that *let-7* is a dose-dependent regeneration suppressor.

Cre under the *Albumin* promoter is expressed in embryonic hepatoblasts that give rise to both hepatocyte and bile duct compartments (*Postic and Magnuson, 1999*, *2000*; *Xu et al., 2006*; *Weisend et al., 2009*; *Malato et al., 2011*), so developmental influences of *let-7* loss could have led to adult phenotypes. To define cell- and temporal-specific roles of *let-7b/c2*, we used adeno-associated virus expressing Cre (AAV8.TBG.PI.Cre.rBG, hereafter called "AAV-Cre"), known to mediate efficient gene excision in hepatocytes but never in biliary epithelial cells (*Yanger et al., 2013*) (*Figure 3—figure supplement 2A,B*). These adult and hepatocyte-specific conditional knockout mice also exhibited significantly enhanced regenerative capacity (*Figure 3—figure supplement 2C–F*). To test if proliferative effects are specific to particular *let-7* species, we knocked-down either *let-7a* or *let-7b* in SV40 immortalized hepatocytes (H2.35 cells) and found that both led to increased proliferation (*Figure 3I*). Collectively, our data shows that physiological *let-7* levels regulate the kinetics of adult liver regeneration by hepatocytes.

## *let-7g* suppresses liver regeneration through insulin-PI3K-mTOR

*let-7* was previously demonstrated to regulate the insulin-PI3K-mTOR pathway (*Zhu et al., 2011*; Frost and Olson, 2011), which is also important in liver regeneration (*Okano et al., 2003*; *Chen et al., 2009*; *Haga et al., 2009*; *Espeillac et al., 2011*). To avoid auto-regulatory feedback and compensation as confounding factors, we focused on liver tissues exposed to acute *let-7* gain or loss. In regenerating livers treated with *let-7g* mimic (*Figure 2I–K*), we found significant protein suppression of insulin receptor β, Igf1rβ, and Irs2, previously validated *let-7* targets at the top of the insulin pathway (*Figure 4A,B*) (*Zhu et al., 2011*). In addition to insulin signaling components, the expression of cell cycle genes (*Ccnb1*, *Cdc34*, and *Cdk8*) and *Map4k4* were also downregulated (*Figure 4C*). In mice with acute *let-7b/c2* deletion by AAV-Cre (*Figure 3—figure supplement 2*), there was a small increase in insulin receptor β protein levels (*Figure 4D,E*). Increased mTOR signaling was also evident in the increased phospho-S6K/Total S6K and phospho-S6/Total S6 ratios (*Figure 5E*).

To determine if mTOR signaling is functionally relevant in LKO mice, we treated mice with rapamycin two hours prior to and immediately after PHx. Rapamycin abrogated differences in regenerating liver weights between control and LKO mice (*Figure 4F*), demonstrating that *let-7b/c2* loss promotes additional mTOR activation to enhance regeneration. Rapamycin's allosteric inhibition of mTOR can lead to pleiotropic and unpredictable effects due to cell-type specific and feedback related phenomena (*Thoreen et al., 2009*). INK128 is a second generation mTOR inhibitor that directly competes with ATP at the catalytic domains of mTORC1/2, leading to more complete abrogation of 4EBP and S6K1 (*Hsieh et al., 2012*). INK128, similar to rapamycin, completely abrogated the regenerative enhancement associated with *let-7b/c2* loss (*Figure 4G*). Analysis of p-S6K confirmed that mTOR is hyperactivated in LKO livers and that INK128 extinguishes the mTOR dependent phosphorylation of this substrate (*Figure 4H*). Similar results after rapamycin and INK128

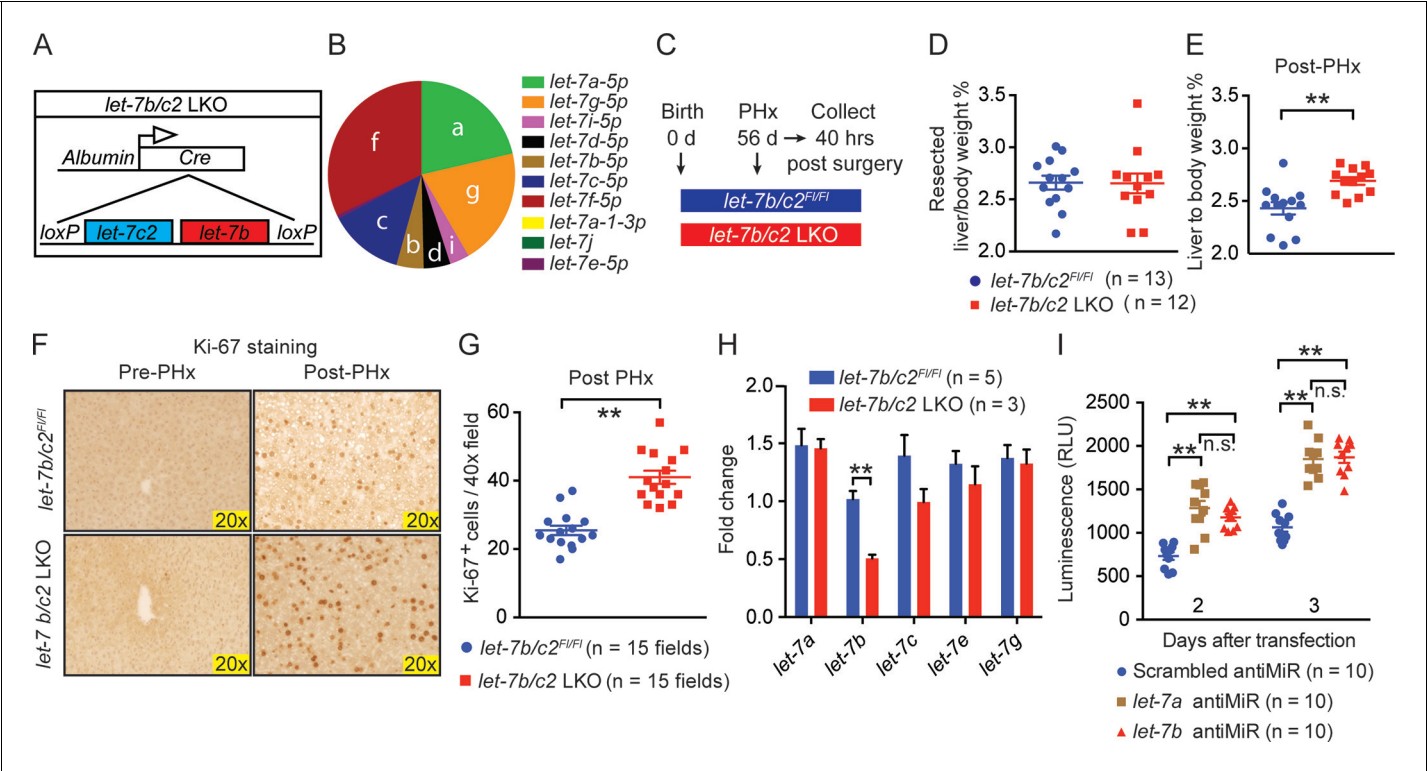

**Figure 3.** Loss of *let-7b* and *let-7c2* is sufficient to enhance liver regeneration. (A) Schema of liver-specific *let-7b* and *let-7c2* knockout mice (*let-7b/c2* LKO). *Albumin-Cre* excises loxPs in the embryonic liver of *let-7b/c2*$^{Fl/Fl}$ mice. Mice without *Cre* serve as the controls. (B) Small RNA sequencing showing the distribution of 10 *let-7s* in WT mice (n=2) (Data obtained from Xie et al. 2012). (C) Schema showing that PHx was performed on *let-7b/c2*$^{Fl/Fl}$ and *let-7b/c2* LKO mice at 56 days of age and tissues were collected 40 hr post PHx. (D) Resected liver/body weight ratios at the time of PHx, and (E) Liver to body ratios of *let-7b/c2*$^{Fl/Fl}$ (n=11) and *let-7b/c2* LKO mice (n=10) 40 hr after PHx. (F) Ki-67 staining and (G) Quantification of Ki-67-positive cells on resected and 40 hr post-PHx liver tissues (n=3 and 3 mice; total of five 40x fields/mouse were used for quantification). (H) RT-qPCR on *let-7* family members from *let-7b/c2*$^{Fl/Fl}$ and *let-7b/c2* LKO mice pre- and 40 hr post-PHx. (I) Viability of H2.35 immortalized human hepatocytes treated with either scrambled, *let-7a*, or *let-7b* antiMiRs, measured at two and three days after transfection (n=10 each). All data in this figure are represented as mean ± SEM. *p<0.05, **p<0.01.

The following figure supplements are available for figure 3:

**Figure supplement 1.** Characterization of *let-7b/c2* LKO mice.

**Figure supplement 2.** Post-natal deletion of *let-7b/c2* also enhances liver regeneration.

indicated that mTOR and its substrates play an essential role in driving increased regeneration in the context of *let-7* suppression.

## Chronic high-dose *let-7g* causes hepatotoxicity and liver carcinogenesis

Since acute *let-7g* induction interferes with hepatocyte proliferation, we asked what the effects of chronic high-dose *let-7g* induction might be. To answer this question, we induced *let-7g* using *rtTA* under the control of the *Rosa* promoter, which drives higher expression than the *LAP* promoter (*Figure 5A*). These mice lost significant body weight and became jaundiced (*Figure 5—figure supplement 1A* and *Figure 5*). Liver function tests indicated severe liver injury (*Figure 5C*) and histology showed prominent microvesicular steatosis, characterized by intra-cytoplasmic lipid droplets (*Figure 5D*), a finding associated with drug-induced liver injury, acute fatty liver of pregnancy, or Reye's syndrome in humans. Using this system, mature *let-7g* was overexpressed by more than twentyfold, as compared to ~eightfold induction in the *LAP-let-7S21L* system (*Figure 5E*). Liver dysfunction was not seen after low-dose *let-7* overexpression in *LAP-let-7S21L* mice (*Figure 2—figure*

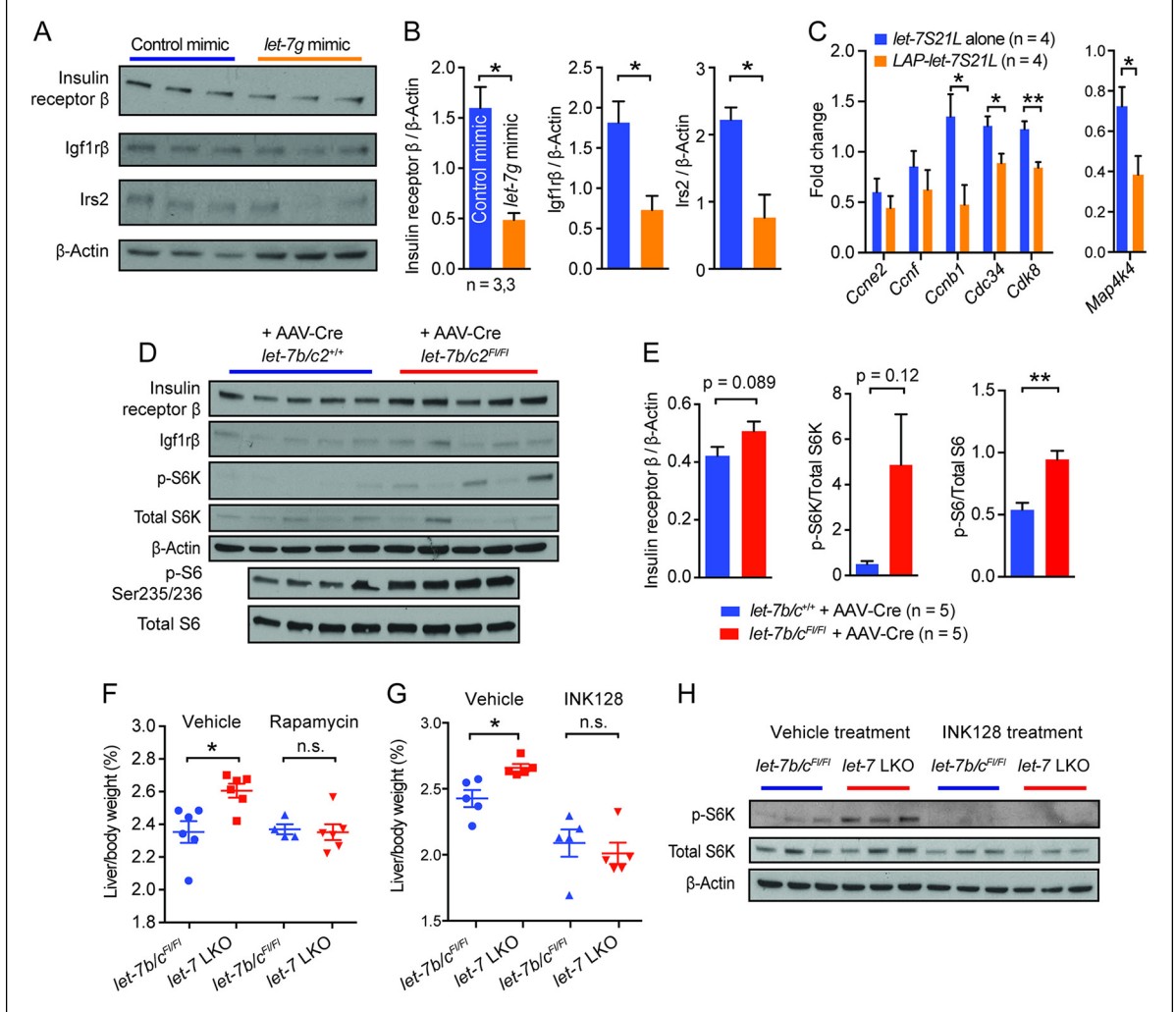

**Figure 4.** *let-7g* suppresses liver regeneration through insulin-PI3K-mTOR. (**A**) Western blots of insulin receptor β, Igf1rβ, Irs2, and β-Actin in negative control or *let-7g* microRNA mimic treated liver tissues 40 hr after PHx. (**B**) Quantification of intensity of insulin receptor β, Igf1rβ, Irs2 (Image J). (**C**) Cell cycle gene expression in *let-7S21L* alone (n=4) and *LAP-let-7S21L* (n=4) livers before and 40 hr after PHx as determined by RT-qPCR. (**D**) Western blots of insulin receptor β, Igf1rβ, p-S6K, total S6K, β-Actin, p-S6 (Ser235/236), and total S6 in AAV-Cre treated *let-7b/c2 +/+* and *let-7b/c2^{Fl/Fl}* livers (n=5 and 5). (**E**) Quantification of intensity of insulin receptor β/β-Actin, p-S6K/total S6K, and p-S6/total S6, 40 hr after PHx (Image J). (**F**) Rapamycin treatment during and after PHx in *let-7b/c2^{Fl/Fl}* control and *let-7b/c2* LKO mice. Shown are liver weights 40 hr post PHx. (**G**) INK128 treatment during and after PHx in *let-7b/c2^{Fl/Fl}* control and *let-7b/c2* LKO mice. Shown are liver weights 40 hr post PHx. (**H**) Western blots of p-S6K, total S6K, and β-Actin in *let-7b/c2^{Fl/Fl}* control and *let-7b/c2* LKO livers treated with either vehicle or INK128 at 40 hr post PHx. All data in this figure are represented as mean ± SEM. *p<0.05, **p<0.01.

supplement 1) or after high-dose *miR-26a-2* overexpression in *Rosa-rtTA; TRE-miR-26a-2* mice (*Figure 5—figure supplement 1B–D*), suggesting a dose and *let-7* microRNA specific effect.

Another possibility was that the *miR-21* loop of the *let-7S21L* construct saturated the microRNA biogenesis machinery, thus causing non-specific toxicity independent of the *let-7* seed sequence. To address this we again delivered a higher dose (2.0 mg/kg) of mature *let-7g* and control microRNA mimics, which do not harbor loops or tails, into wild-type mice. Mice receiving *let-7g* mimic lost significantly more weight (*Figure 5F*) and suffered hepatocyte destruction leading to increased AST/ALT levels (*Figure 5G*), while control mice remained healthy. Mimic delivery achieved a twelvefold increase of *let-7g* (*Figure 5H*). These results suggest that above certain doses *let-7* is incompatible with hepatocyte survival, and that *let-7's* anti-proliferative activities would interfere with normal tissue homeostasis.

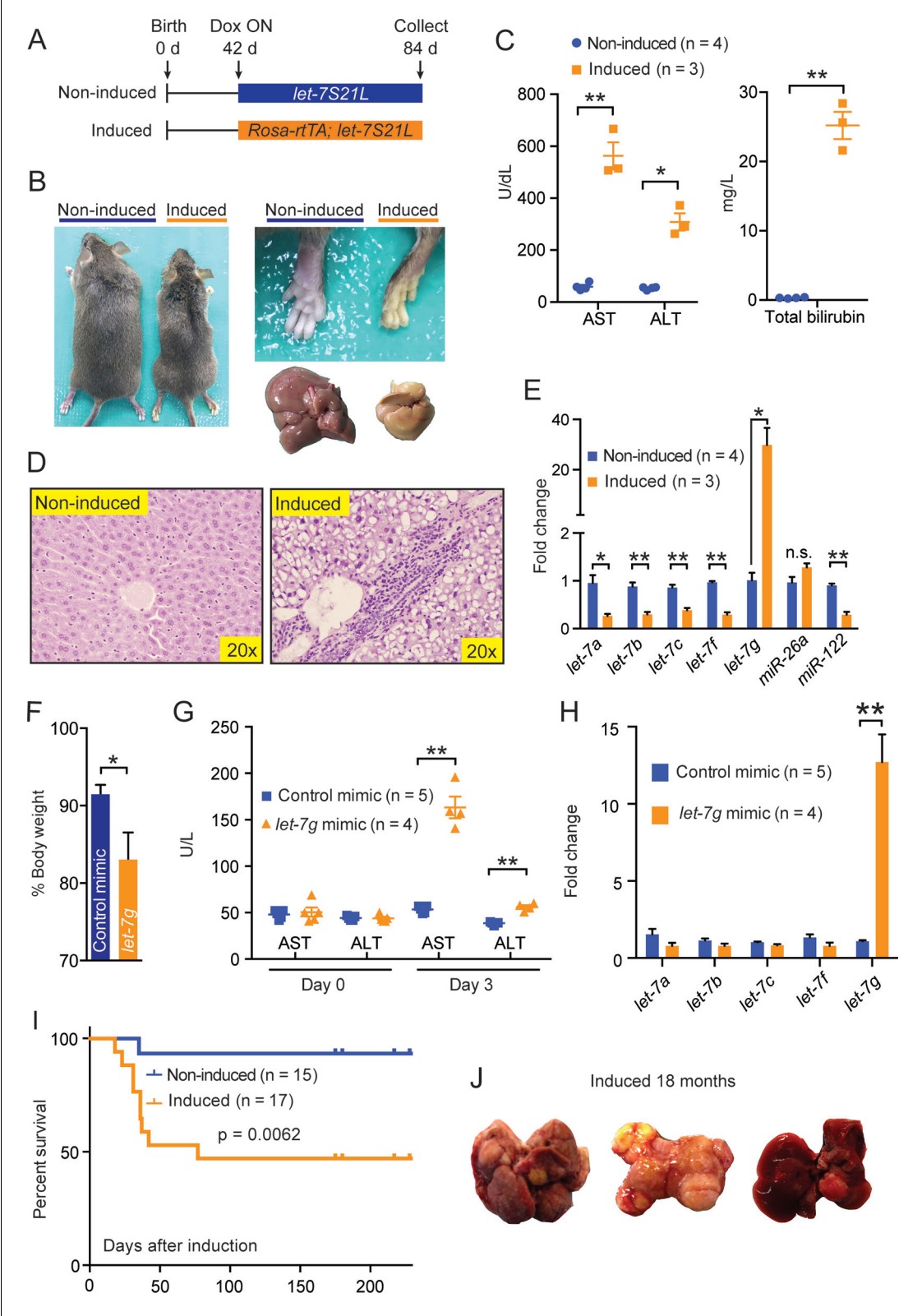

**Figure 5.** Chronic high-dose *let-7g* causes hepatotoxicity and liver carcinogenesis. (**A**) Schema showing that *let-7S21L* control and *Rosa-rtTA; let-7S21L* mice were induced at 42 days of age and collected at 84 days. (**B**) Images showing the whole body, extremities, and livers of *Rosa-rtTA* (n=4) and *Rosa-let-7S21L* mice (n=3) given 1 mg/mL dox between 42 and 84 days of age. (**C**) Liver function tests: AST (U/L), ALT (U/L), and total bilirubin (mg/L) in these mice. (**D**) H&E staining of livers. (**E**) RT-qPCR of mature *let-7s* and other microRNAs in *let-7g* overexpressing mice (n=4 and 3). (**F**) Body weight 3 days

*Figure 5 continued on next page*

*Figure 5 continued*

after injection of 2.0 mg/kg negative control or *let-7*g microRNA mimics packaged in C12-200 LNPs relative to pre-injection weight (n=5 and 4). (**G**) Liver function tests: AST (U/L) and ALT (U/L) in WT C57Bl/6 mice before and 3 days after mimic injection (n=5 and 4). (**H**) Mature *let-7* levels in wild-type C57Bl/6 mice treated with mimics as determined by RT-qPCR (n=5 and 4). (**I**) Kaplan-Meier curve for *Rosa-let-7S21L* induced with 1.0 g/L dox at 6 weeks old (n=15 and 17). (**J**) Gross images of the liver of *Rosa-let-7S21L* mice induced for 18 months. All data in this figure are represented as mean ± SEM. *p<0.05, **p<0.01.

The following figure supplement is available for figure 5:

**Figure supplement 1.** Data associated with Figure 5.

When the *Rosa-let-7S21L* mice were induced chronically, approximately 50% of the mice survived the acute liver injury seen after dox induction (**Figure 5I**). Over the course of 18 months, 5 of 10 (50%) of these surviving mice developed large liver tumors, whereas only 1 of 12 (7.7%) of the non-induced mice had any tumors (**Figure 5J**). Chronic *let-7* overexpression likely caused hepatocyte toxicity and selected for pre-malignant hepatocytes that eventually become cancer. Our long-term experiments revealed the potential dangers of chronic *let-7* treatment, and the consequent disruption of the balance between tissue regeneration, degeneration, and cancer risk.

## Discussion

The role of *let-7*s in adult animal physiology is unclear in part because the redundancy of this large microRNA family has made loss of function studies challenging. Deep redundancy of multiple highly conserved genes raises the possibility that dose regulation is important. Despite this, overexpression has been helpful in uncovering physiological functions of *let-7* (*Zhu et al., 2011*; *Frost and Olson, 2011*; *Shyh-Chang et al., 2013*; *Nishino et al., 2013*; *Madison et al., 2013*). Using overexpression tools, we have shown that *let-7* suppression is a fundamental requirement for *MYC*-mediated liver transformation, and that *let-7* is capable of counteracting strong oncogenic drivers in vivo. However, one negative consequence of raising the level of *let-7* expression is a limitation in the ability to regenerate after major tissue loss. More surprisingly, our knockout mouse model showed that the loss of two out of the ten *let-7* members in the liver resulted in improved liver proliferation and regeneration. These data suggest a lack of complete redundancy between *let-7* microRNA species, but rather a precisely regulated cumulative dose that when increased or decreased, leads to significant alterations in regenerative capacity.

The knowledge that *let-7* suppresses both normal and malignant growth will have particular relevance to malignancies that arise from chronically injured tissues. In these tissues, winners of the competition between cancer and host cells might ultimately dictate whether organ failure or tumor progression ensues. It has been thought that one key advantage of using microRNAs therapeutically is that they are already expressed at high levels in normal tissues, thus making increased dosing likely to be safe and tolerable. Surprisingly, we showed that chronic *let-7* overexpression caused hepatotoxicity, disrupted tissue homeostasis, ultimately leading to carcinogenesis. This is likely due to the high levels of overexpression achieved in the *Rosa-rtTA* transgenic system, as opposed to the lower dose in the *LAP-tTA* system. These high doses are likely to be toxic to hepatocytes, a phenomena compounded by the fact that excess *let-7* impairs proliferation in surviving cells that might serve to replenish lost tissues. Eventually, certain clones must epigenetically or genetically evolve to evade *let-7* growth inhibition in order to transform.

It is also interesting that *let-7* overexpression led to dramatically different outcomes in distinct cancer contexts. While dose is most likely the critical variable between the *Rosa-rtTA* and *LAP-tTA* systems, *Rosa-rtTA* does induce expression in cells other than hepatocytes and bile duct epithelia, leaving the possibility that non-cell autonomous influences of *let-7* overexpression play a role in liver injury and cancer development. A more interesting possibility would be if distinct genetic subtypes of cancer respond differently to *let-7* overexpression. Since *let-7* has been conceptualized as a general tumor suppressor, it is surprising that it can cause opposing phenotypes in distinct cancer models. *MYC* liver cancers show a dramatic suppression of *let-7*, rendering it especially sensitive to *let-7* replacement. Tumors or tissues with more normal levels of *let-7* might not respond to increases in

*let-7*. Alternatively, the growth of other cancer models may not depend on the overproduction of *let-7* target genes/proteins. *let-7* overexpression in these contexts would probably not elicit growth suppression, but may instead exacerbate tissue injury. It would be interesting to evaluate the effects of *let-7* overexpression in hepatocellular carcinomas caused by different driver mutations. Together, our data suggest that *let-7* therapy directed at hepatocellular carcinomas could be risky, given that most of these cancers occur in severely compromised, cirrhotic livers (*Yang et al., 2011*).

We speculate that the total dose of *let-7* is evolutionarily determined via regulation of the expression levels of individual *let-7* members, and is postnatally maintained at a level that can suppress cancer, but which also allows for adequate levels of mammalian regenerative capacity. Clearly, *let-7* levels are not static throughout life, since *let-7* levels are dynamic after environmental perturbations. However, when baseline *let-7* levels are altered permanently by genetic means, compromises in tumor suppression or tissue regeneration were revealed. Our study underscores the importance of regulating appropriate levels of this small RNA to maintain health during times of regenerative stress.

## Materials and methods

### Mice
All mice were handled in accordance with the guidelines of the Institutional Animal Care and Use Committee at UTSW. *MYC* tumor models and the *LAP-let-7S21L* inducible mice were carried on a 1:1 FVB/C57Bl/6 strain background. Please see (*Nishino et al., 2013*) for more details about the *let-7b/c2* floxed mice, which are on a C57Bl/6 background. The chronically injured *let-7* inducible mice were on a mixed B6/129 background. All experiments were done in an age and sex controlled fashion unless otherwise noted in the figure legends.

### Partial hepatectomy
Two-thirds of the liver was surgically excised as previously described (*Mitchell and Willenbring, 2008*).

### RNA extraction and RT-qPCR
Total RNA was isolated using Trizol reagent (Invitrogen). For RT-qPCR of mRNAs, cDNA synthesis was performed with 1 ug of total RNA using miScript II Reverse Transcription Kit (Cat. #218161, Qiagen). Gene expression levels were measured using the ΔΔCt method as described previously (*Zhu et al., 2010*).

### Western blot assay
Mouse liver tissues were ground with a pestle and lysed in T-PER Tissue Protein Extraction Reagent (Thermo Scientific Pierce). Western blots were performed in the standard fashion. The following antibodies were used: Anti-Insulin receptor β (Cell Signaling #3025), Anti-Igf1rβ (Cell Signaling #9750), Anti-Irs2 (Cell Signaling #3089), Anti-total S6K (Cell Signaling #9202), Anti-p-S6K (Cell Signaling #9205), Anti-total S6 (Cell Signaling #2217), Anti-p-S6 Ser235/236 (Cell Signaling #2211), Anti-mouse β-Actin (Cell Signaling #4970), Anti-rabbit IgG, HRP-linked Antibody (Cell Signaling #7074) and Anti-mouse IgG, HRP-linked Antibody (Cell Signaling #7076).

### Histology and immunohistochemistry
Tissue samples were fixed in 10% neutral buffered formalin or 4% paraformaldehyde (PFA) and embedded in paraffin. In some cases, frozen sections were made. Immunohistochemistry was performed as previously described (*Zhu et al., 2010*). Primary antibodies used: Ki-67 (Abcam #ab15580). Detection was performed with the Elite ABC Kit and DAB Substrate (Vector Laboratories), followed by Hematoxylin Solution counterstaining (Sigma).

### Liver function tests
Blood samples (~50 ul) were taken retro-orbitally in heparinized tubes. Liver function tests were analyzed by the UTSW Molecular Genetics core.

## Viral Cre excision

100 µL of AAV8.TBG.PI.Cre.rBG (University of Pennsylvania Vector Core) was retro-orbitally injected at a titer of 5 x$\times$ $10^{10}$ genomic particles to mediate 90%-–100% Cre excision.

## Cell culture and in vitro antiMiR experiments

The H2.35 cell line was directly obtained from ATCC and has been cultured for less than 6 months. The cells were authenticated by ATCC using Short Tandem Repeat (STR) DNA profiling. Cells were cultured in DMEM with 4% (vol/vol) FBS, 1x Pen/Strep (Thermo Scientific) and 200 nM Dexamethasone (Sigma). Cells were transfected with control (Life Technologies Cat. AM17010), *let-7a* (Life Technologies Cat. #4464084-Assay ID MH10050), or *let-7g* (Life Technologies Cat. #4464084-Assay ID MH11050) miRVana antiMiRs. AntiMiRs were packaged by RNAiMAX (Invitrogen) and transfected into H2.35 cells cultured in 96-well plates at a concentration of 50 nM. The number of viable cells in each well was measured at 2 and 3 days after transfection using CellTiter-Glo Luminescent Cell Viability Assay (Promega Cat. #G7570).

## In vivo microRNA mimic experiments

For in vivo experiments, formulated C12-200 lipidoid nanoparticles (LNPs) were used to package either Control (Life Technologies Cat. #4464061) or *let-7g* (Life Technologies Cat. 364 #4464070-Assay ID MC11758) miRVana mimic at either 0.5 or 2 mg/kg and delivered intravenously through the tail vein. LNPs were formulated following the previously reported component ratios (*Love et al., 2010*) with the aid of a microfluidic rapid mixing instrument (Precision Nanosystems NanoAssemblr) and purified by dialysis in sterile PBS before injection.

## In vivo drug treatments

Rapamycin (LC Biochem) was dissolved in 25% ethanol/PBS and then injected at 1.5 mg/kg 2 hr prior to and 20 hr after PHx. INK128 (LC Biochem) was formulated in 5% polyvinylpropyline, 15% NMP, 80% water and administered by oral gavage at 1 mg/kg 2 hr prior to and 20 hr after PHx.

## MicroRNA sequencing

Female CD1 mice were treated with 3 mg/kg TCPOBOP in DMSO-corn oil by gavage (*Tian et al., 2011*), sacrificed at 3, 6, 9, 12, and 18 hr after treatment, and compared to untreated controls. Replicate libraries were made from two individual mice for each condition. RNA was purified with the Qiagen miRNeasy Mini kit. Small RNA libraries were constructed using an Illumina Truseq Small RNA Sample Prep Kit. 12 indexed libraries were multiplexed in a single flow cell lane and received 50 base single-end sequencing on an Illumina HiSeq 2500 sequencer. Sequence reads were aligned to mm9 using Tophat and quantified with Cufflinks by the FPKM method (*Trapnell et al., 2012*). Data for each experimental condition represent the average values from two libraries.

## Statistical analysis

The data in most figure panels reflect multiple experiments performed on different days using mice derived from different litters. Variation is always indicated using standard error presented as mean ± SEM. Two-tailed Student's *t*-tests (two-sample equal variance) were used to test the significance of differences between two groups. Statistical significance is displayed as $p<0.05$ (*) or $p<0.01$ (**) unless specified otherwise. In all experiments, no mice were excluded form analysis after the experiment was initiated. Image analysis for the quantification of cell proliferation, cell death, and fibrosis were blinded.

## Acknowledgements

LW was supported by The Science and Technology Program of Guangzhou, China (2012J5100031) and Youth Teachers Cultivation Project of Sun Yat-Sen University (12ykpy21). LHN was supported by the Howard Hughes Medical Institute (HHMI) Pre-doctoral International Student Fellowship. JTM was supported by the Cancer Prevention and Research Institute of Texas (CPRIT grant R1008) and NIH (R01CA120185). SJM is an HHMI Investigator, the Mary McDermott Cook Chair in Pediatric Genetics, the director of the Hamon Laboratory for Stem Cells and Cancer, and a CPRIT Scholar.

SJM was also supported by a grant from the National Institute on Aging (AG024945). GQD is an investigator of HHMI and the Manton Center for Orphan Disease Research and was supported by NIGMS R01GM107536. DJS acknowledges support from CPRIT (grant R1212) and the Welch Foundation (I-1855). NSC was supported by a NSS Scholarship from the Agency for Science, Technology and Research, Singapore. HZ was supported by a American Cancer Society Postdoctoral Fellowship, a NIH/NCI K08 grant (5K08CA157727), a NIH/NCI R01 grant (1R01CA190525), the Pollack Foundation, a Burroughs Welcome Career Medical Award, and a CPRIT Recruitment Award (R1209).

## Additional information

### Competing interests

SJM: Senior editor, *eLife*. The other authors declare that no competing interests exist.

### Funding

| Funder | Grant reference number | Author |
|---|---|---|
| National Cancer Institute | 1K08CA157727-02 | Hao Zhu |
| Howard Hughes Medical Institute | | Sean J Morrison<br>George Q Daley |
| Howard Hughes Medical Institute | 59108180 | Liem H Nguyen |
| Cancer Prevention and Research Institute of Texas | R1209 | Hao Zhu |
| National Cancer Institute | 1R01CA190525-01 | Hao Zhu |
| National Institute of General Medical Sciences | R01GM107536 | George Q Daley |
| The Science and Technology Program of Guangzhou, China | 2012J5100031 | Linwei Wu |
| National Institute on Aging | AG024945 | Sean J Morrison |
| Welch Foundation | I-1855 | Daniel J Siegwart |
| Agency for Science, Technology and Research | NSS Scholarship | Ng Shyh-Chang |
| Youth Teachers Cultivation Project of Sun Yat-Sen University | 12ykpy21 | Linwei Wu |
| Burroughs Wellcome Fund | Career Award for Medical Scientists | Hao Zhu |
| Pollock Foundation | | Hao Zhu |
| Cancer Prevention and Research Institute of Texas | | Joshua T Mendell<br>Sean J Morrison |
| Cancer Prevention and Research Institute of Texas | R1212 | Daniel J Siegwart |
| American Cancer Society | Postdoctoral Fellowship | Hao Zhu |
| Cancer Prevention and Research Institute of Texas | R1008 | Joshua T Mendell |
| Cancer Prevention and Research Institute of Texa | R01CA120185 | Joshua T Mendell |

The funders had no role in study design, data collection and interpretation, or the decision to submit the work for publication.

### Author contributions

LW, LHN, JT, JL, JTM, HZ, Conception and design, Acquisition of data, Analysis and interpretation of data, Drafting or revising the article; KZ, Conception and design, Acquisition of data, Contributed unpublished essential data or reagents; TYdeS, DJS, GQD, NSC, Conception and design, Acquisition

of data, Analysis and interpretation of data, Drafting or revising the article, Contributed unpublished essential data or reagents; LL, Acquisition of data, Analysis and interpretation of data, Contributed unpublished essential data or reagents; JBM, GS, MTS, Conception and design, Acquisition of data, Analysis and interpretation of data, Contributed unpublished essential data or reagents; SZ, LRZ, AA, Conception and design, Acquisition of data, Analysis and interpretation of data; SCW, Conception and design, Drafting or revising the article; XH, Analysis and interpretation of data, Drafting or revising the article; JN, Analysis and interpretation of data, Contributed unpublished essential data or reagents; SJM, Drafting or revising the article, Contributed unpublished essential data or reagents

### Author ORCIDs

Daniel J Siegwart, http://orcid.org/0000-0003-3823-1931

### Ethics

Animal experimentation: This study was performed in strict accordance with the recommendations in the Guide for the Care and Use of Laboratory Animals of the National Institutes of Health. All of the animals were handled according to approved institutional animal care and use committee (IACUC) protocols (#2012-0143) of the University of Texas Southwestern Medical Center. All surgery was performed under isoflurane anesthesia with appropriate analgesia, and every effort was made to minimize suffering.

## Additional files

### Major datasets

The following datasets were generated:

| Author(s) | Year | Dataset title | Dataset URL | Database, license, and accessibility information |
|---|---|---|---|---|
| Xie J, Ameres SL, Friedline R, Hung JH et al | 2011 | AAV vector-mediated in vivo miRNA antagonism for studying miRNA function and treating dyslipidemia | http://www.ncbi.nlm.nih.gov/geo/query/acc.cgi?acc=GSE25971 | Publicly available at the NCBI Gene Expression Omnibus (Accession no: GSE25971). |

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
