## [Decision Letter]

Thank you for submitting your work entitled "Precise *Let-7* expression levels balance organ regeneration against tumor suppression" for peer review at *eLife*. Your submission has been favorably evaluated by Fiona Watt (Senior editor), a Reviewing editor, and two reviewers.

The following individual responsible for the peer review of your submission has agreed to reveal his identity: Frank Slack (peer reviewer). A further reviewer remains anonymous.

The reviewers have discussed the reviews with one another and the Reviewing editor has drafted this decision to help you prepare a revised submission.

Summary:

In the manuscript entitled "Precise *Let-7* expression levels balance organ regeneration against tumor suppression" Wu and colleagues utilize transgenic overexpression and liver-specific knockout of *Let-7* to assess the consequences of *Let-7* dysregulation in the setting of *MYC*-driven carcinogenesis and during liver regeneration after partial hepatectomy. The authors find that moderate *Let-7* overexpression blocks MYC-driven hepatoblastoma formation and inhibits liver regeneration after partial hepatectomy. During liver regeneration, *Let-7* overexpression alters steady state levels of various metabolites. Further, *Let-7* regulates the insulin-mTOR pathway after regeneration. Finally, extreme *Let-7* overexpression causes liver injury and liver carcinogenesis.

All reviewers found this work to be innovative and important, noting particularly the likely impact on the field of demonstrating the functional role of *Let-7* regulation during carcinogenesis and liver repair and the potential negative effects of high dose *Let-7* on normal liver function. However, two reviewers expressed significant concerns about the rather expansive scope of the manuscript in its current form, which causes the work to appear somewhat superficial and difficult to follow in places. The authors are strongly encouraged to consider how they may address this issue in their revision, perhaps by separating some of the multiple, intertwined stories or by providing clearer connections between the various observations that are reported.

Essential revisions:

1) Please revise the manuscript to provide more focus on the major conclusions and message and a clearer, more streamlined description of the experiments and data.

2) Some conclusions presented appear to be stronger than others, and the paper would benefit from inclusion of relevant experimental caveats. For example, the partial hepatectomy studies only examine one time point, at 40 hours post resection, thereby only investigating the initial phase of hepatocyte proliferation, rather than also the subsequent phase of vascular-driven liver remodeling. It is possible that *Let-7* is more dynamically regulated and active in both phases. Similar concerns apply to the metabolomics studies, which, while innovative, are problematic in that they represent a snap shot of the metabolic state at one time point. They provide no assessment of metabolite flux, which would require more extensive labeling studies. Further, the increase in nucleotides, for example, could well be due to increased biosynthesis, but could also be due to a block in nucleotide utilization. If rapid cell proliferation is required for liver regeneration, requiring rapid DNA synthesis, would DNA assembly be the rate-limiting step, causing accumulation of nucleotides? Also, would it be informative to correlate metabolite changes in the glycolysis or nucleotide biosynthesis pathways with expression changes of relevant enzymes? Please address these issues experimentally or by inclusion of relevant discussion and caveats.

3) The role of mTOR in liver regeneration has been described, and the connection to the *Let-7* pathway is intriguing, as is the fact that rapamycin decreases liver recovery. However, the pleiotropic effects of rapamycin are also well-described, and should be noted. The use of more specific agents (e.g. Torin) or genetic methods, if available, might also be desirable for a more comprehensive assessment.

4) Please comment on whether differences in cellular context (i.e., oncogenetic background) in addition to or instead of *Let7g* levels may explain the different outcomes in the *myc*-driven vs. spontaneous tumor models.

---

## [Author Response]

*All reviewers found this work to be innovative and important, noting particularly the likely impact on the field of demonstrating the functional role of* Let-7 *regulation during carcinogenesis and liver repair and the potential negative effects of high dose* Let-7 *on normal liver function. However, two reviewers expressed significant concerns about the rather expansive scope of the manuscript in its current form, which causes the work to appear somewhat superficial and difficult to follow in places. The authors are strongly encouraged to consider how they may address this issue in their revision, perhaps by separating some of the multiple, intertwined stories or by providing clearer connections between the various observations that are reported. Essential revisions: 1) Please revise the manuscript to provide more focus on the major conclusions and message and a clearer, more streamlined description of the experiments and data.*

We have revised the paper so that it is clearer and more streamlined. Redundant data was removed. For example, the experiment involving *MYC* +/- *let-7S21L* induction at 4 weeks as opposed to 2 weeks of age, which revealed similar findings, was removed. The metabolism data was removed as suggested by the editors and reviewers. The language in many sections was also shortened and clarified.

*2) Some conclusions presented appear to be stronger than others, and the paper would benefit from inclusion of relevant experimental caveats. For example, the partial hepatectomy studies only examine one time point, at 40 hours post resection, thereby only investigating the initial phase of hepatocyte proliferation, rather than also the subsequent phase of vascular-driven liver remodeling. It is possible that* Let-7 *is more dynamically regulated and active in both phases.*

It is possible that *let-7* is dynamically regulated and important during multiple mechanistically independent phases of liver regeneration, although it is unlikely given the *let-7* expression data. Figure 2 shows that *let-7* was significantly altered only during the period shortly after hepatectomy and reverted back to baseline levels at around 40 hours.

Vascular-driven liver remodeling was previously shown to be influential 4-7 days after partial hepatectomy (Ding et al. Nature 2010). To completely exclude the possibility that this mechanism is influenced by *let-7*, we assessed the liver regeneration at 4, 7 and 14 days post partial hepatectomy. In both loss and gain of function models, there were no significant differences in regenerated liver mass (Figure 2—figure supplement 1 and Figure 3—figure supplement 1). These results support the idea that *let-7* expression changes lead to alterations in the initial, but not subsequent phases of liver regeneration after partial hepatectomy.

*Similar concerns apply to the metabolomics studies, which, while innovative, are problematic in that they represent a snap shot of the metabolic state at one time point. They provide no assessment of metabolite flux, which would require more extensive labeling studies. Further, the increase in nucleotides, for example, could well be due to increased biosynthesis, but could also be due to a block in nucleotide utilization. If rapid cell proliferation is required for liver regeneration, requiring rapid DNA synthesis, would DNA assembly be the rate-limiting step, causing accumulation of nucleotides? Also, would it be informative to correlate metabolite changes in the glycolysis or nucleotide biosynthesis pathways with expression changes of relevant enzymes? Please address these issues experimentally or by inclusion of relevant discussion and caveats.*

As suggested by the editors and reviewers, we excluded the metabolomics data from this manuscript in an effort to focus on the major findings of the paper.

*3) The role of mTOR in liver regeneration has been described, and the connection to the* Let-7 *pathway is intriguing, as is the fact that rapamycin decreases liver recovery. However, the pleiotropic effects of rapamycin are also well-described, and should be noted. The use of more specific agents (e.g. Torin) or genetic methods, if available, might also be desirable for a more comprehensive assessment.*

Rapamycin has pleiotropic and cell-type dependent effects. This is partly because it acts allosterically through FKBP12 rather than mTOR itself, and thus does not completely inhibit mTORC1 and mTORC2. As a result, variable inhibition of mTORC1 substrates can lead to incomplete protein synthesis inhibition via p-4EBP. In contrast, second generation ATP-competitive mTOR kinase inhibitors such as Torin and INK128 directly antagonize mTORC1/2 complexes and thus are more potent and specific inhibitors (Hsieh et al., 2012).

To confirm the rapamycin results with a second compound, and to examine the impact of more potent mTOR blockade, we performed partial hepatectomy in *let-7b/c* LKO mice treated with INK128. Torin1 was not used in part because of poor mouse microsome stability and a short *in vivo* half-life (Liu and Gray et al., Bioorg Med Chem Lett. 2011), and because we have had previous successes using INK128 in vivo.

INK128, similar to rapamycin, completely abrogated the regenerative enhancement associated with *let-7* loss (Figure 4). p-S6K western blots confirmed that mTOR is hyperactivated in LKO livers and that INK128 extinguishes mTOR activity (Figure 4). Identical results after rapamycin and INK128 indicated that the mTOR signaling pathway and its substrates (especially S6K) play an essential role in driving increased regeneration in the context of *let-7* suppression.

*4) Please comment on whether differences in cellular context (i.e., oncogenetic background) in addition to or instead of* Let7g *levels may explain the different outcomes in the* myc*-driven vs. spontaneous tumor models.*

Additional comments/clarification from the reviewers:

*The authors use 2 different tumor models to assess the effects of* let-7 *overexpression on tumorigenesis in the liver, and they observe two different outcomes. In the current manuscript, they attribute the difference in outcome solely to differences in expression levels of* let-7 *(8 fold in one model and 20-fold in the other). However, there are many other differences in the oncogenetic backgrounds of the models that also could contribute to or account for the differences. For instance, in one case where* let-7 *reduces tumor burden, tumors arise due to strong induction of oncogenic* Myc*; these tumors have a 3-5 month latency. In the other case, tumors arise spontaneously, with unknown oncogenic mutations, and show an 18 mo. latency. Thus, an alternative interpretation of the data is that* let-7 *overexpression in* Myc *overexpressing tumors is tumor suppressive whereas* let-7 *overexpression in tumors lacking* Myc *overexpression is tumor promoting.*

*This would be analogous to observations in* Drosophila *indicating that oncogenic Ras converts TNF signaling from anti- to pro-oncogenic (see, e.g., Cordero et al. Dev. Cell 18:999, 2010). Or, it may be that* let-7 *overexpression in young mice (<6 mo) is tumor suppressive whereas* let-7 *overexpression in aged mice (>12mo.), which have acquired additional age-associated genetic, epigenetic and microenvironmental alterations, is tumor promoting. One could test this experimentally, by using the* RosaLet7S21L *system in the* LAP-tTA; TRE-Myc *background, but as this would be a rather time-consuming experiment, the authors may instead wish simply to discuss more clearly the other differences in the models that might contribute to the different outcomes, rather than attributing outcomes only to* let-7 *expression levels.*

We agree that cellular context or oncogenetic background in addition to *let-7* dose may indeed contribute to the different cancer outcomes. We did not to make the claim that *let-7* overexpression is effective in *MYC* cancers solely because it is expressed at low doses. We only stated that *let-7* dose is a critically important variable in liver injury (in the absence of *MYC*). As a key piece of evidence, we directly compared low and high dose *let-7* overexpression in normal livers. Long-term low dose *let-7* overexpression in *LAP-let-7S21L* mice (Figure 2 and Figure 2—figure supplement 1) never resulted in liver dysfunction or tumor formation. This was emphasized and clarified in the text.

Age should also be excluded as a possibility, since high dose *let-7* was induced by *Rosa-rtTa* from 2 or 4 weeks onward, mirroring the *MYC + let-7* induction times. There are however caveats to consider. The *LAP* and *Rosa* promoters induce expression in different cells at different levels, resulting in complex non-cell autonomous influences that might play a role in liver cancer development. We have commented on this possibility in the revised paper.

Certainly, *MYC* tumors may be particularly sensitive to *let-7* overexpression because of its specific oncogenetic context. *MYC*-driven liver cancers show a dramatic suppression of *let-7*; other tumor or tissue contexts may not share this extent of *let-7* suppression. It would make sense that this context is especially sensitive to *let-7* replacement, whereas tumors or tissues with normal levels of *let-7* might not respond to *let-7* increases. Alternatively, other cancer models may not depend on the overproduction of *let-7* target genes/proteins. Overexpressing *let-7* in those contexts would probably not suppress growth, but may instead exacerbate tissue injury. We have not had time to thoroughly evaluate other mouse models of liver cancer, although this would clearly be interesting.

Unfortunately, we have not had time to test high and low dose *let-7* expression in the context of *MYC* liver cancers. This is a difficult experiment for our transgenic systems because dose titration of *let-7* will also influence the dose of *MYC* (both are controlled by tet sensitive promoters), which is known to also influence tumor biology.

We have included a paragraph in the Discussion stating that it is possible that oncogenetic differences underlie the ways in which *let-7* impacts cancer biology.